# Covariance matrix filtering with bootstrapped hierarchies

**Christian Bongiorno** [ORCID] [^*], **Damien Challet** [^]

Université Paris-Saclay, CentraleSupélec, Mathématiques et Informatique pour la Complexité et les Systèmes, Gif-sur-Yvette, France

[^] These authors contributed equally to this work.
* christian.bongiorno@centralesupelec.fr

**Data Availability Statement:** Financial Data cannot be shared publicly although they are publicly available online. We included in the electronic supplementary material the code that we used to download the data from Yahoo Finance.

## Abstract

Cleaning covariance matrices is a highly non-trivial problem, yet of central importance in the statistical inference of dependence between objects. We propose here a probabilistic hierarchical clustering method, named Bootstrapped Average Hierarchical Clustering (BAHC), that is particularly effective in the high-dimensional case, i.e., when there are more objects than features. When applied to DNA microarray, our method yields distinct hierarchical structures that cannot be accounted for by usual hierarchical clustering. We then use global minimum-variance risk management to test our method and find that BAHC leads to significantly smaller realized risk compared to state-of-the-art linear and nonlinear filtering methods in the high-dimensional case. Spectral decomposition shows that BAHC better captures the persistence of the dependence structure between asset price returns in the calibration and the test periods.

## Introduction

Covariance matrix estimation is a cornerstone of dependence inference between objects. Unfortunately, this kind of matrix becomes very noisy when the number of objects is similar to the number of features, a phenomenon known as the curse of dimensionality. Even worse, unfiltered covariance matrices are pathological when the number of features exceeds the number of objects, i.e., in the so-called high dimensional case. This case is frequent e.g. in biological data and in multivariate dynamical systems such as financial markets in which only the most recent history is likely to be relevant.

Given its importance, covariance matrix filtering has a long history. A popular approach is to obtain a filtered covariance matrix from the corresponding correlation matrix. Two types of approaches stand out: *i)* spectral methods, e.g. Random Matrix Theory, Rotationally Invariant Estimators [1], and Shrinkage [2, 3]; *ii)* ansatz for the correlation matrix, e.g. block-diagonal [4] or hierarchical [5].

The usual setting is to have $n$ objects and $t$ features and to compute the correlation matrix between these $n$ objects. Recent results on Rotationally Invariant Estimators [6] propose algorithms able to correct the eigenvalue spectrum of covariance matrices optimally without filtering its eigenvectors: the inversion of the QuEST function [7], the Cross-Validated (CV)

**Funding:** The author(s) received no specific funding for this work.

**Competing interests:** The authors have declared that no competing interests exist.

eigenvalue shrinkage [8] and the IW-regularization [1], the latter being valid only in the low dimensional regime $q = n/t < 1$, i.e., when there are more features than objects. Direct eigenvector filtering is more complex. An indirect way to filter both eigenvectors and eigenvalues is to use ansätze for the shape of the true correlation matrix, which also impose constraints on the structure of both the eigenvectors and the eigenvalues. A good ansatz should be simple enough to clean noise but flexible enough to account for fine relevant details. The popular hierarchical clustering ansatz (HC thereafter) is indeed simple: it assumes that correlations are nested [5, 9], which is equivalent to assume that dependencies are described by a dendrogram (a tree).

An obvious problem of HC occurs when the structure is more complex than a tree: for example, the non-diagonal blocks in Figs 1 and 2 are erased by a hierarchical ansatz. As a consequence, a non-negligible part of the dependence structure is left out. In these cases, the tree inferred by a hierarchical ansatz is fragile with respect to small data perturbations such as bootstraps. The fragility itself was noted for example in Ref. [10] which showed that only a subset of links of a minimum spanning tree associated to a HC are reliable when data are perturbed by bootstraps. In practice, it is hard to find statistically-validated hierarchical structures [11] when the fitted hierarchical structure is highly sensitive to small variations of data.

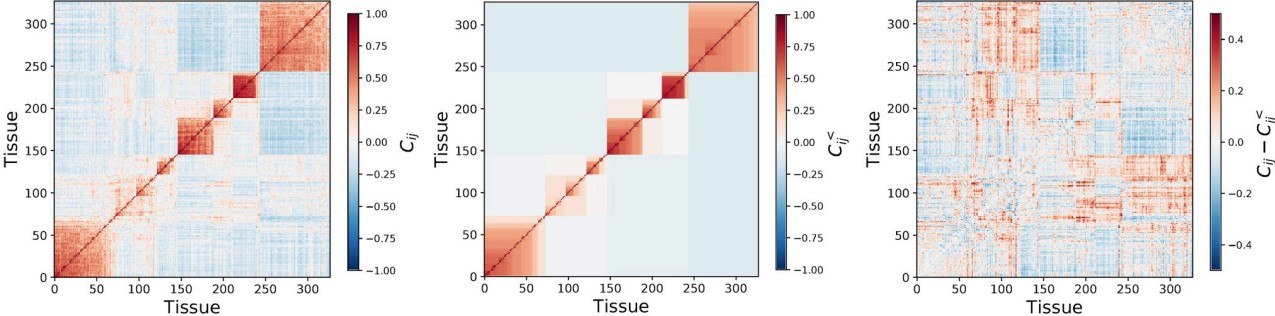

**Fig 1. Correlation matrix from tissue-gene micro-array data of patients affected by lung cancer.** The upper left plot is the sample correlation matrix, the upper right plot is the result of hierarchical and average-linkage averaging (HCAL). The bottom left plot is the difference between the two: it still has evident structure unaccounted for by HCAL.

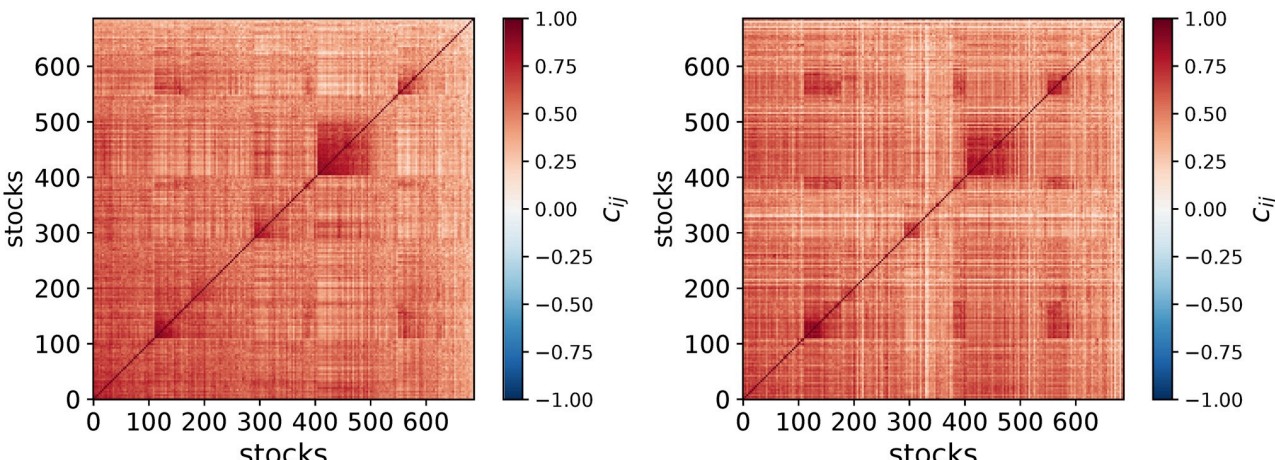

**Fig 2. Correlation matrix of US equities price returns in the 2008-01-23 to 2008-11-04 (left plot) and in the 2008-11-05 to 2009-08-24 period (right plot).** The elements of both panels are ordered according to the in-sample HCAL dendrogram of the first period.

Here, we introduce a more flexible method able to capture more of the structure of the eigenvectors. The idea is to create many bootstrapped copies of the original data and to apply hierarchical clustering average linkage (HCAL) [5] filtering to each of them. We then average all these HCAL-filtered matrices. We call our method BAHC, which stands for Bootstrapped Average Hierarchical Clustering, and define it for covariance and correlation matrices. A BAHC-filtered matrix is a sum of multiple hierarchical structures weighted by their frequency. A single hierarchical structure will only emerge if all the bootstrap realizations lead to the same dendrogram. Thus, this method is particularly adapted to data that is well-described by a hierarchical structure in a first approximation [12] but avoids selecting a single fragile structure.

We illustrate the power of our method with data from two relevant fields. First, in bioinformatics, DNA micro-array gene expression dependence in tissues is frequently characterized by correlation matrices. Hierarchical clustering and its variants are commonly used [13, 14], which helps simplify the covariance matrix by linkage averaging [15] (see Fig 1). When there are several different candidates of hierarchical structure, this approach only selects a single one, which neglects possibly crucial information held by alternative structures. Comparing unfiltered correlation matrices with the filtering yielded by hierarchical clustering and average linkage (HCAL) [5] (Fig 1) makes it clear first that (i) hierarchical clustering does capture some of the structure and (ii) a substantial part of the structure is lost (see the bottom plot). This is because hierarchical clustering imposes too strict a structure, which erases out an uncontrolled amount of information.

Another domain in which covariance matrix filtering plays a central role is risk management in many areas. Broadly speaking, the problem amounts to minimize future uncertainty by determining the fraction of resources to allocate to every possible choice. Risk in this particular context is due to fluctuations of the future value of the choices. The usual procedure consists in minimizing a suitable risk measure in the calibration window and hoping that the future, realized, risk will bear some relationship with the calibrated risk.

The simplest approach consists in defining risk as the variance of the weighted sum of choices' values and to minimise it. This is known as global minimum-variance portfolios, a subfield of quadratic portfolio optimization which has a wide range of applications: investment into technologies [16], energy sources mix for countries [17, 18], wind farm locations [19], and capital allocation in finance [20]. We shall focus on financial risk because data are abundant, which makes it possible to compare the out-of-sample performance of filtering methods. In addition, the high-dimensional regime is particularly relevant in finance: there are many assets to choose from and the speed with which the dependence structure between asset price returns may change asks for an as short as possible calibration period [21].

In an inference or descriptive context such as DNA microarray data analysis, filtering correlation matrices is meant to bring estimated covariance matrices closer to the ground truth. In a dynamical context, especially for non-stationary systems such as financial markets, what matters is the part of the ground truth that most likely persists after the calibration period, i.e., when one uses the allocation weights computed from the filtered covariance matrix. Thus, ideally, the filtered covariance matrix should contain as much of the persistent structure as possible. The nature of the most likely persistent structure is of course unknown from the calibration window only. Fig 2 shows that there are indeed strongly persistent dependence structures of asset price returns between two non-overlapping periods. Similarly to correlation matrices of DNA microarray data, while a pure HC does capture a sizeable part of the useful structure, the non-diagonal correlation patterns blocks e.g., around $(x, y) = (140, 600)$ indicate that HC itself is not sufficient.

## Methods

### Datasets description

We consider the daily close-to-close returns from 1992-02-03 to 2020-03-31 of US equities, adjusted for dividends, splits, and other corporate events. More precisely, the dataset consists of 1295 assets taken from the union of all the components of the Russell 1000 from 2010-06 to 2020-03. The number of stocks with data varies over time: it ranges from 151 in 1992-06-22 to 1172 in 2018-01-17 (see S1 File for a code to download the data).

DNA microarray data [22] can be downloaded from [23]. It consists of gene expression intensity of 327 tissues of patients affected by pediatric acute lymphoblastic leukemia and a subset of 271 genes.

### Numerical simulations with financial data

All the simulations are carried out in the same way: each point of each plot is an average over 10, 000 simulations, each of which includes an in-sample window of length $t_{in}$ and an out-of-sample window of length $t^{out} = 42$ days (about two trading months) unless otherwise specified; it starts from a random day uniformly chosen in the available dataset. To have meaningful in- and out-of-sample windows given the maximum $t^{in}$ considered, the first day of the out-of-sample must be after 01-01-2000; each simulation selects $n = 100$ assets at random among the assets with no missing value in both in- and out-of-sample windows.

### BAHC algorithm

Given matrix $R \in \mathbb{R}^{n \times t}$, our method prescribes to create a set of $m$ (feature-wise) bootstrap copies of $R$, denoted by $\{R^{(1)}, R^{(2)}, \cdots, R^{(m)}\}$. A single bootstrap copy of the data matrix $R^{(b)} \in \mathbb{R}^{n \times t}$ has elements $r_{ij}^{(b)} = r_{is_j^{(b)}}$, where $s^{(b)}$ is a vector of dimension $t$ obtained by random sampling with replacement of the elements of vector $\{1, 2, \cdots, t\}$. The vectors $s^{(b)}$, $b = 1, \cdots, m$ are independently sampled.

The Pearson correlation matrix of each bootstrapped data matrix $R^{(b)}$ is then computed and denoted by $C^{(b)}$; in turn the latter is filtered with the hierarchical clustering average linkage (HCAL) proposed in [5], which yields $C^{(b)<}$. In short, HCAL uses two ingredients: the distance $D = 1 - C$ to agglomerate cluster in a hierarchical way, and the averaging of the correlation between clusters (see S1 Appendix for more details).

Finally, the filtered correlation matrix $C^{\text{BAHC}}$ is the average of the HCAL-filtered matrices $C^{(b)<}$

$$C^{\text{BAHC}} = \frac{1}{m} \sum_{b=1}^{m} C^{(b)<}.$$

To build a BAHC-filtered covariance matrix, we estimate the standard deviation of $r_i$, denoted by $\sigma_{ii}$, and obtain the element of the BAHC-filtered covariance matrix as

$$\sigma_{ij}^{\text{BAHC}} = c_{ij}^{\text{BAHC}} \sqrt{\sigma_{ii} \sigma_{jj}}.$$

**Source code.** We have written a BAHC package for both R and Python, available from CRAN and PyPI, respectively.

## Frobenius norms

We use rescaled Frobenius norms to account for the fact that the number of assets in our data-set depends on time, defined as

$$\| X \|_F^\Sigma = \sqrt{\sum_{i,j}^{n \times n} \frac{x_{ij}^2}{n^2}}. \tag{1}$$

In addition, because CV, LW and QuEST methods do not guarantee the identity on the diagonal of filtered correlation matrices (contrarily to BAHC), we do not include the diagonal elements in the metric and thus define

$$\| X \|_F^C = \sqrt{\sum_{i>j}^{n \times n} \frac{2 \, x_{ij}^2}{n(n-1)}}. \tag{2}$$

We found that the performance of CV, LW, QuEST-based correlation estimators is slightly improved by replacing $c_{ij}$ with $\frac{c_{ij}}{\sqrt{c_{ii} \, c_{jj}}}$, which also ensures that the diagonal elements equal one, and thus have used this modification in our analysis.

## Results

### Microarray DNA

We first apply the BAHC method to DNA microarray data [22] where the objects are $n = 327$ tissues of patients affected by pediatric acute lymphoblastic leukemia and features are the expression intensities of $t = 271$ genes ($q \simeq 1.21$). Classifying leukemia subtypes based on their gene expression profile is crucial to correct prognosis and risk assessment. However, the simplistic classification obtained from a single tree could lose relevant information coming from more complex dependence structures.

To show the new insights brought by BAHC compared to a simple hierarchical clustering, we kept the dendrograms of all the bootstraps used to compute the BAHC-filtered correlation matrix and produced a bidimensional t-SNE projection [24] using the pairwise cophenetic correlation coefficient as a distance. In this map, each point corresponds to a bootstrapped copy of the original data. Two such copies are represented nearby if the cophenetic correlation between their HC-filtered dendrogram is high—in simple words, if they are similar. If two randomly chosen bootstrap dendrograms differ only due to sample size error, we should expect such bi-dimensional mapping scattered around an average dendrogram. However, two main clusters of dendrograms appear. They essentially differ by the topmost branches, as shown by the tanglegram of the centroids of these two clusters (right plot of Fig 3). This means that in this dataset, a small perturbation not only affects the lower levels of the dendrograms, whose composition is based on the stability single or pairs of correlation coefficients that are necessarily highly affected by sample size error, but also the highest aggregate levels, which should be more robust to sample size noise. In other words, the appearance of two clear clusters of dendrograms shows that a single dendrogram fails to account for the real dependence between gene expression intensity. In addition, clades that are distant on the sample dendrogram may be much closer in both of these clusters.

This shows that even a large distance between two sub-groups of elements (cancers, in this case) may not be stable with respect to small perturbation of the data. Thus, if one wishes to cluster genes, one should generate bootstrapped dendrograms and then apply a clustering

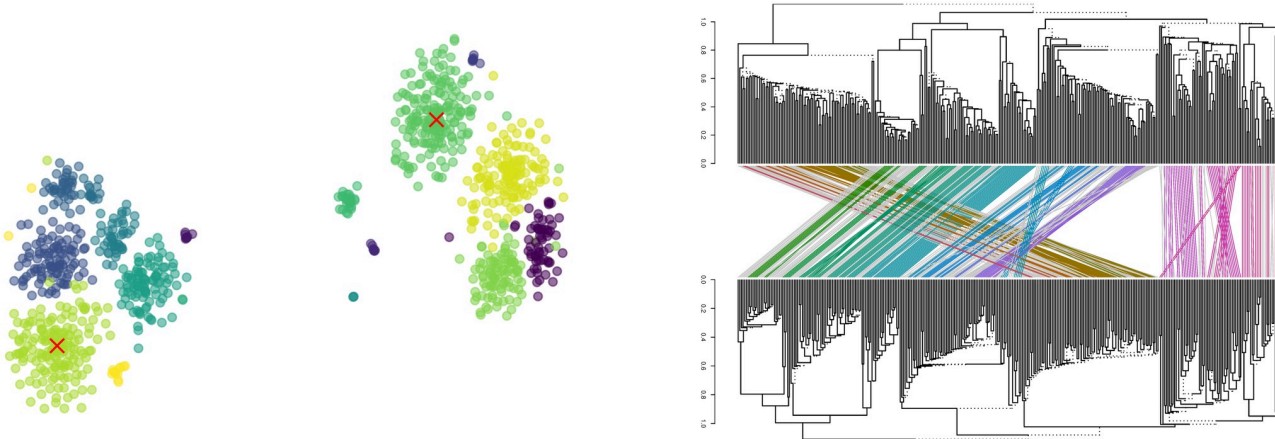

**Fig 3. Bidimensional t-SNE projection of the cophenetic distance between the dendrograms of 1000 bootstraps of DNA microarray data [22].** Two main clusters emerge, with further subclusters, corresponding to distinct potential hierarchies of dependence that are compatible with data. The red crosses indicate the centroids of the two largest clusters whose structure differences appear in the tanglegram of right plot.

method adapted to trees, as we did above. If one needs a filtered covariance matrix, one should use BAHC instead of a HC.

## Risk minimization

Given the $n \times (t + 1)$ matrix of values of choice $i$ at time $k$, $p_{i,k}$, and the value returns $r_{i,k} = p_{i,k}/p_{i,k-1} - 1$, one must determine the fraction of investment given to each choice $i$, the $i$-th component of vector $\mathbf{w}$. The risk is measured by the standard deviation of the portfolio return, denoted by $v_P$, with $v_P^2 = \mathbf{w}^T \Sigma \mathbf{w}$, where $\Sigma$ is the $n \times n$ covariance matrix of the matrix of returns $R$. If the weights can be negative, the optimal weights $\tilde{\mathbf{w}} = \frac{\Sigma^{-1} \cdot \mathbb{1}}{\mathbb{1}^T \cdot \Sigma^{-1} \cdot \mathbb{1}}$, with the condition $\Sigma_i w_i = 1$ in order to avoid the trivial solution $\mathbf{w} = 0$. This situation is called long-short portfolio in the following. In some situations, e.g., when choosing one's portfolio of energies or products, only positive weights are allowed, in which case one has to solve a quadratic programming problem; we refer to this situation as long-only portfolio.

The realized (out-of-sample) risk is the relevant performance measure. Using the $^{out}$ exponent, the realized risk is

$$v_P^{out} = \sqrt{(\tilde{\mathbf{w}})^{\dagger} \Sigma^{out} \tilde{\mathbf{w}}},$$

where $\tilde{\mathbf{w}}$ are computed from the in-sample covariance matrix, filtered or not, and $X^{\dagger}$ is the transpose of matrix $X$.

All the results reported below use the simulation setup described in the Methods section: in short, we perform 10,000 simulations of $n = 100$ random assets in random periods. We compare the out-of-sample risk computed from BAHC and several other well-known methods: the classic Ledoit and Wolf linear shrinkage method (LW henceforth) [2] and the more recent nonlinear shrinkage approach based on the inversion of the QuEST function (QuEST) [7]. We also include the Cross-Validated eigenvalue shrinkage (CV) [8] and HCAL [5], denoted by $<$.

Fig 4 shows that BAHC outperforms all the alternative methods for $t^{in} \lesssim 200$, i.e., for $q = n/t \gtrsim \frac{1}{2}$, which includes all of the high-dimensional regime $q > 1$. In particular, for the long-only portfolios, the BAHC method reaches the absolute minimum out-of-sample risk over all $t^{in}$ and all methods for $t^{in} \simeq 200$, i.e., $q \simeq 1/2$. The right-hand-side plots of Fig 4 report the probability that BAHC outperforms each alternative method when $q > 1/2$, which

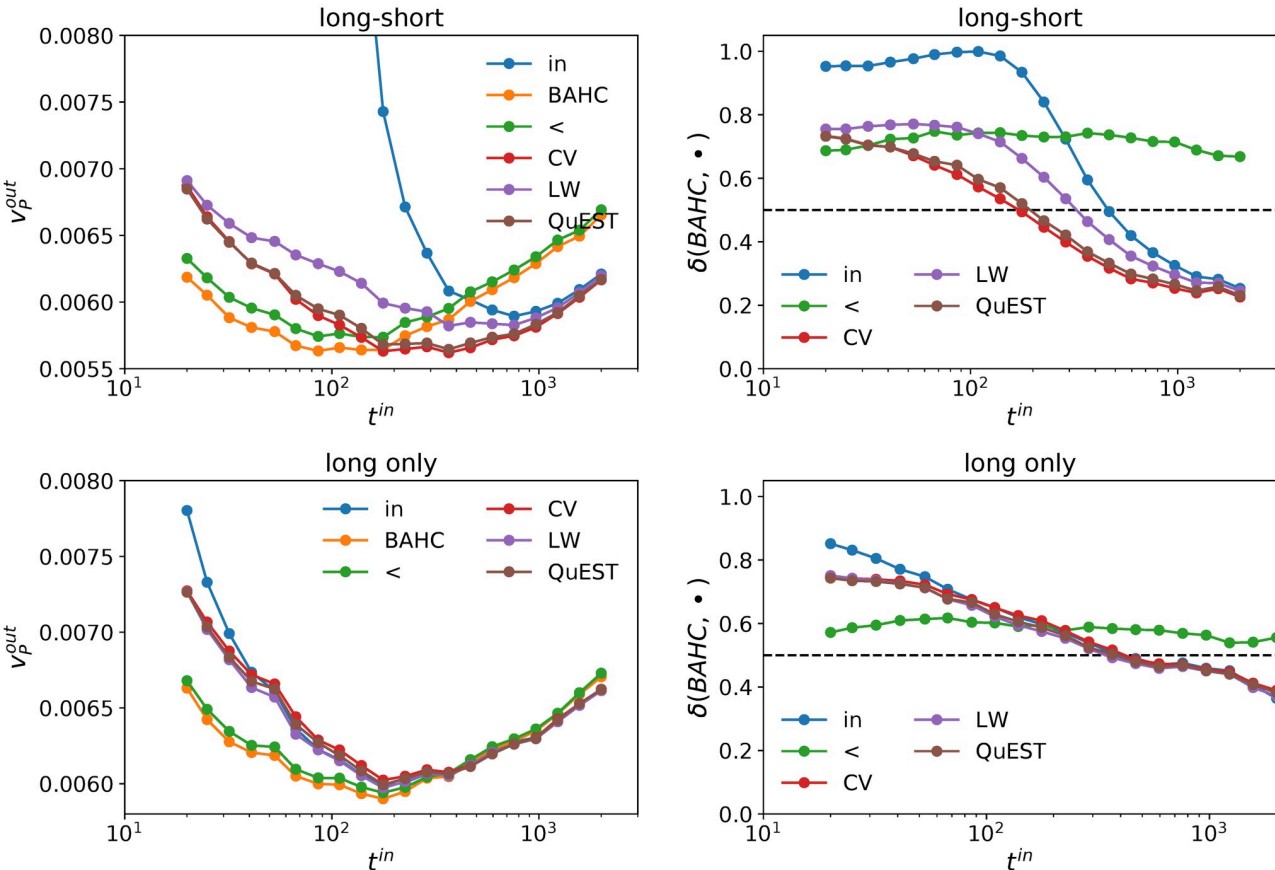

**Fig 4. Left plots: Realized risk for different estimators; right plots: Fraction of time the realized risk of BAHC is smaller than the one obtained with alternative estimators.** 10, 000 independent simulations per point; $t^{out} = 42$ days, $n = 100$ assets, US equities.

confirms that BAHC is better than all the other methods not only with respect to the average realized risk, but also in probability in this region.

Finally, we vary the length of the test window, $t^{out}$. We report the probability that the BAHC method outperforms all its competitors as a function of both $t^{in}$ and $t^{out}$ in Fig 5. Our approach achieves lower realized risk with in more than half the simulations than any other method tested here as soon as $t^{in} < 177$ ($q > 1/1.17$) for every $t^{out}$ in the considered range. Remarkably, as $t^{out}$ increases, the calibration length below which BAHC has better than 50% chances to outperform all its competitors only weakly increases. We interpret this result by the fact that our method is able to extract the right kind of persistent structure in that particular data, which is confirmed below by spectral analysis. We found similar results for the Hong Kong equity market (see S1 Appendix). We also report in the S1 Appendix an alternative analysis where the out-of-sample standard deviations are used to compute the portfolio compositions. This analysis aims to isolate the effect of correlation filtering approaches providing a lower bound for risk minimization. However, we did not observe any qualitative differences.

## Spectral properties

In order to understand why and when our method has a better performance than the other methods based on spectral clustering, it is instructive to compare the in- and out-of-sample persistence of the eigenvalues and eigenvectors produced by all the filtering methods

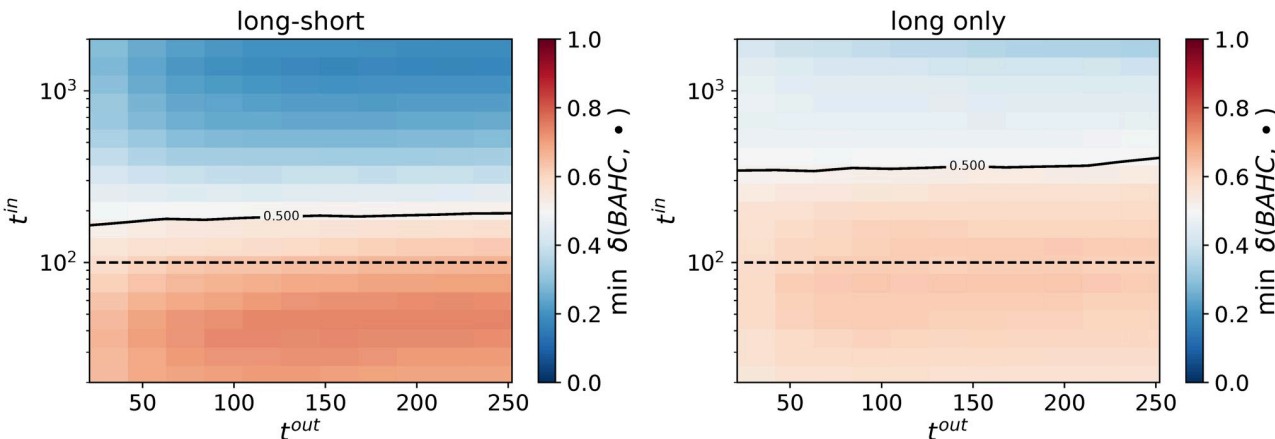

**Fig 5. Fraction of time BAHC yields a smaller realized risk than all the alternative methods.** Left plot: portfolios with positive and negative weights; right plot: portfolios with only positive weights. The dotted line corresponds to $q = t/n = 1$, and the level curve to a 50% probability. 10, 000 independent simulations per point; $t^{out} = 42$ days, $n = 100$ assets, US equities.

considered here. The spectral decomposition of correlation matrix $C$ is denoted by $C = U^\dagger \Lambda U$, where $U$ is a $n \times n$ matrix formed by the eigenvectors of $C$ and $\Lambda$ is the diagonal matrix obtained from the corresponding eigenvalues.

**Eigenvectors stability.** A simple way to characterise the overall eigenvectors stability is to compare the empirical out-of-sample correlation matrix $C^{out}$ with the Oracle correlation estimator defined as $\Xi_C^{in} = U^{in\dagger} Z^{in} U^{in}$ where $Z^{in} = \mathrm{diag}(U^{in\dagger} C^{out} U^{in})$ is the Oracle eigenvector estimator, the idea being that $\Xi_C^{in} = C^{out}$ if the in- and out-of-sample eigenvectors coincide (see S1 Appendix). The Oracle estimator for the covariance matrix, denoted by $\Xi_\Sigma^{in}$, is defined in a similar way.

Fig 6 reports the Frobenius distances (see the Methods section) $\| C^{out} - \Xi_C^{in} \|_F^C$ and $\| \Sigma^{out} - \Xi_\Sigma^{in} \|_F^\Sigma$ as a function of $t^{in}$ for $n = 100$ assets. Note that CV, LW and QuEST methods all use the in-sample eigenvectors and thus we do not need to report separate results. Generally, our method yields more stable correlation and covariance matrices not only in the high-dimensional case, but also up to ($q \simeq 3$), *i.e.* $t^{in} < 300$. The difference is due to the fact that the eigenvectors obtained by our method are more stable than the vanilla in-sample eigenvectors, which mechanically improves the Oracle estimator.

Fig 6 also shows that the probability that the eigenvectors of BAHC-filtered correlation matrices are more stable than those provided by the alternative filtering methods grows as $t^{in}$ becomes smaller. The same applies to the comparison between BAHC -filtered and empirical covariance matrices, while HCAL, denoted by $<$, has better performance in about a 20% of samples almost independently of $t^{in}$. In short, as soon as $q > 1/3$ in this dataset, the BAHC method likely yields more persistent eigenvectors than all the other filtering methods considered here.

**Eigenvalues stability.** Since both the covariance $\Sigma$ and precision $\Sigma^{-1}$ matrices are relevant to minimum-variance optimization, we measure two types of residues that focus on large and small eigenvalues, defined as

$$\epsilon_{hi} = \sqrt{\frac{1}{n} \sum_{i=1}^{n} (\lambda_i - z_i)^2} \tag{3}$$

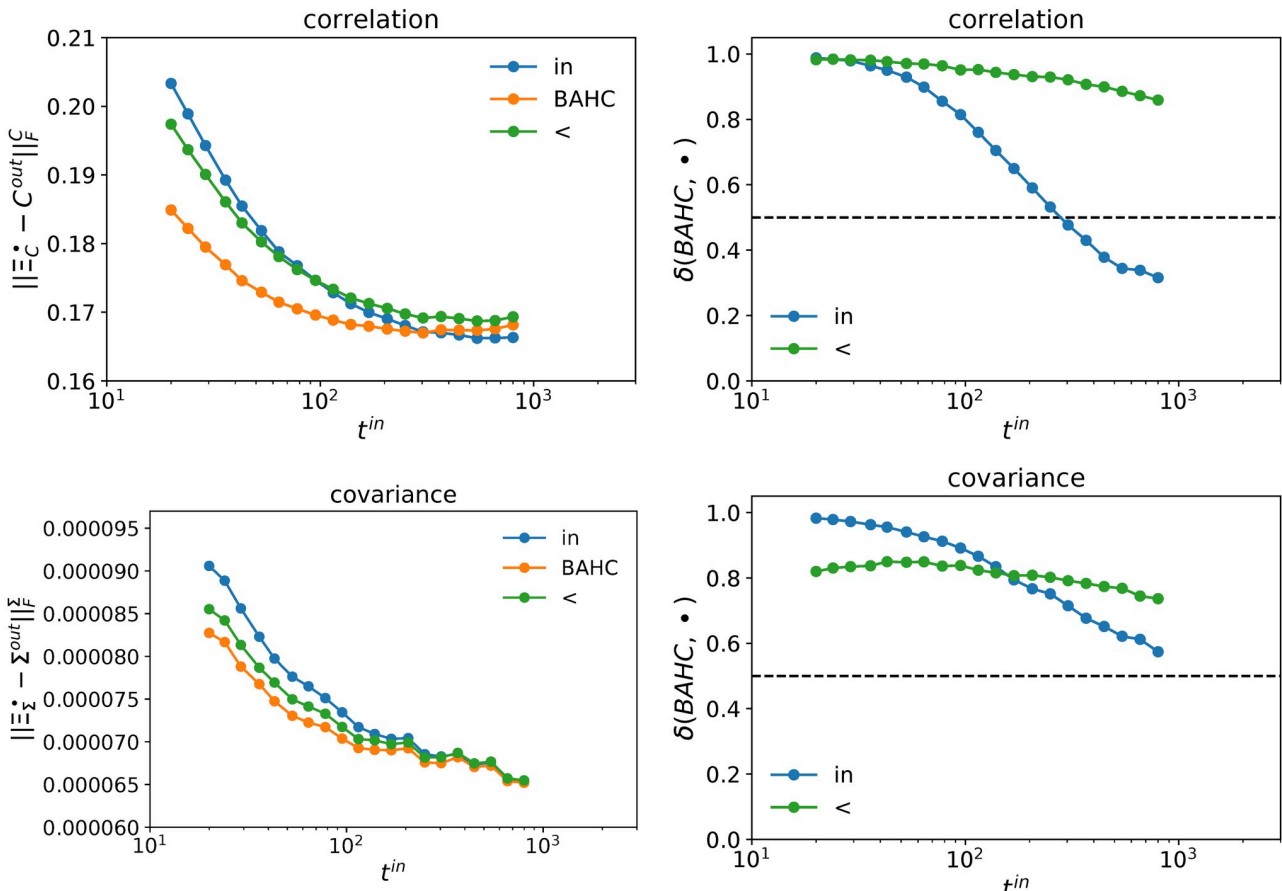

**Fig 6. Frobenius distance between the out-of-sample matrices and the Oracle estimators obtained with the in-sample eigenvectors (*in*), the in-sample BAHC-filtered eigenvectors (*BAHC*) and the in-sample HCAL-filtered eigenvectors (<).** Upper panels refer to correlation matrices $C$, lower panels to covariance matrices $\Sigma$. The left panels are the Frobenius norm of the difference between the estimator and the out-of-sample realization; the right panels are the fraction of time BAHC outperforms the alternative estimators. 10, 000 independent simulations per point; $t^{out} = 42$ days, $n = 100$ assets, US equities.

$$\epsilon_{low} = \sqrt{\frac{1}{n}\sum_{i=1}^{n}\left(\frac{1}{\lambda_i} - \frac{1}{z_i}\right)^2}, \tag{4}$$

where $\lambda_i = (\Lambda)_{ii}$ is the $i$-th (ranked) eigenvalue of the in-sample estimator and $z_i = (Z^{in})_{ii}$ comes from the Oracle estimator computed with the respective filtered eigenvector matrix and $i$ is the respective rank of these eigenvalues. The residue measure $\epsilon_{hi}$ mainly accounts for the discrepancy between the largest eigenvalues and the residue measure $\epsilon_{low}$ attributes more weight to the discrepancy between the smallest eigenvalues.

Fig 7 plots the residues of the correlation and covariance matrices respectively as a function of $t^{in}$. We compare our approach with the sample estimator, HCAL-filtered matrix, and the Cross-Validated (CV) eigenvalue distribution. While CV method outperforms all the other methods when $t^{in} \lesssim 1000$ ($q > 0.01$), the eigenvalues produced by our method are still much closer to the Oracle than those of the raw sample estimator when $t^{in} \lesssim 500$.

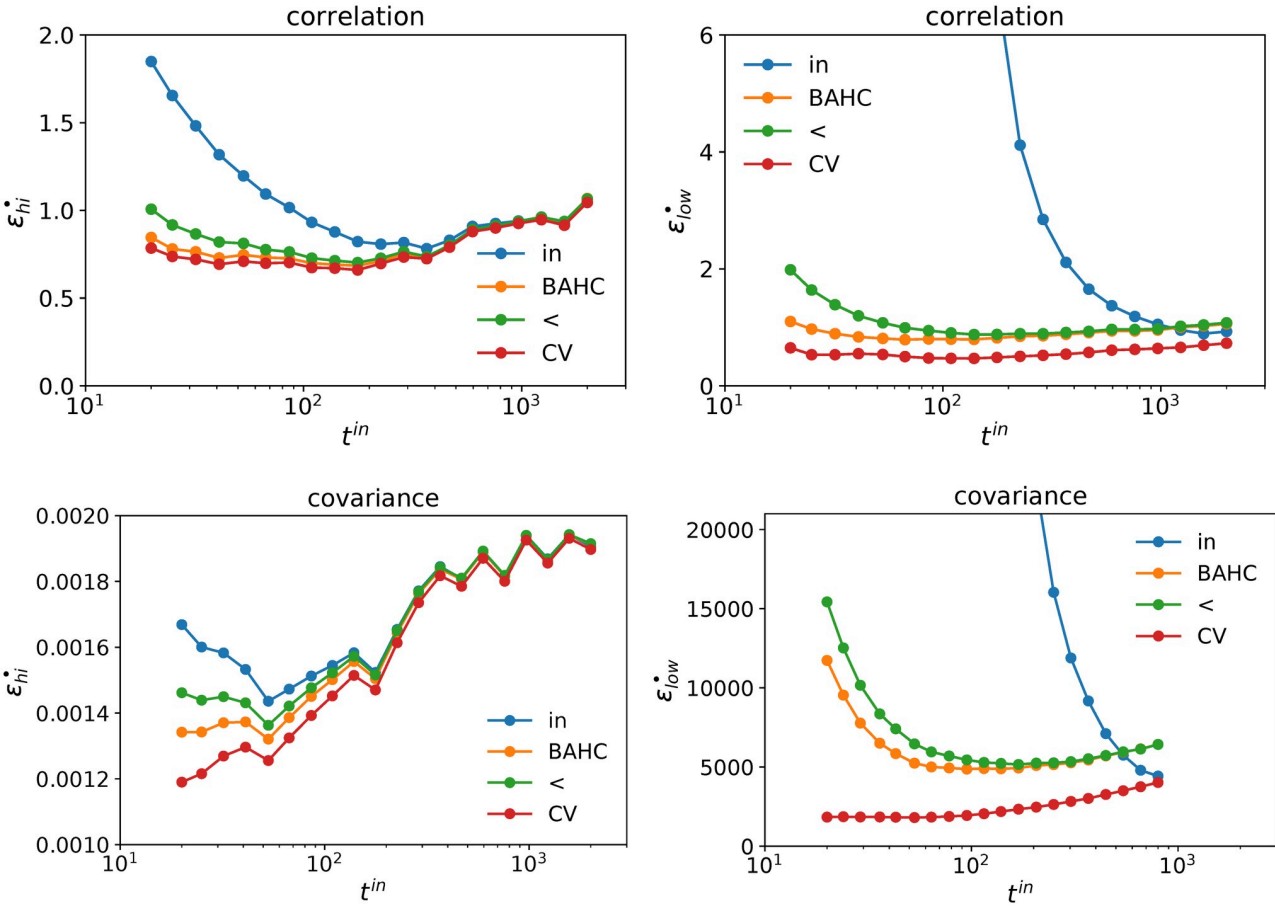

**Fig 7. Average residue $\epsilon_{hi}$ and $\epsilon_{low}$ over 10, 000 simulations with random calibration windows and a random selection of $n$ = 100 assets.** The upper panel refers to the correlation matrix, the lower panel refers to the covariance matrix. 10, 000 independent simulations per point; $t^{out}$ = 42 days, $n$ = 100 assets, US equities.

### Filtered correlation and covariance matrices

The ultimate test is of course to compare filtered in-sample matrices with out-of-sample matrices. Fig 8 reports the Frobenius distance between the filtered in-sample and out-of-sample correlation and covariance matrices for all the tested methods. Expectedly, BAHC outperforms all the other ones for $t^{in} \lesssim 300$. Fig 8 plots the fraction of times the Frobenius norm of our method is lower than the other methods, which confirms the superiority of BAHC for $q \leq 2$ and also shows that BAHC method HCAL filtering for every $t^{in}$. Once again, this emphasizes that a strict hierarchical structure is not sufficient to capture the stable structure of eigenvectors fully.

## Conclusions

Filtering covariance and correlation matrices requires to take care of $O(n^2)$ coefficients. Focusing on $O(n)$ variables, for example by tweaking the eigenvalues or using a single hierarchical ansatz, works to some extend. Making further progresses requires to filter more variables, if possible while keeping an $O(n)$ ansatz. This is what the BAHC method achieves: by using $m$ bootstraps and applying an $O(n)$ structure, BAHC allows some additional flexibility, while keeping the overall structure simple.

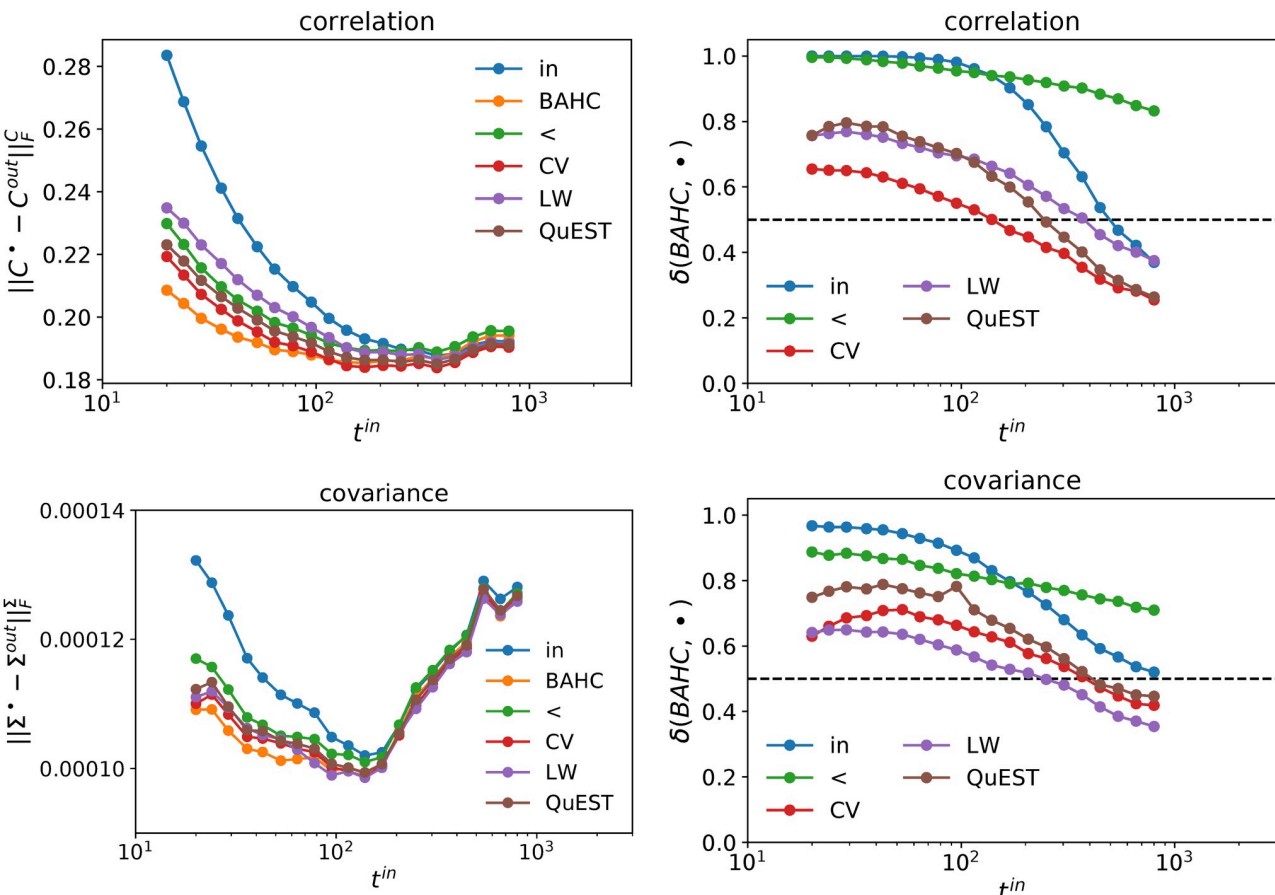

**Fig 8.** Left plots: Frobenius distance between out-of-sample matrices and filtered in-sample matrices; upper panels refer to correlation matrices $C$, lower panels to covariance matrices $\Sigma$. Right plots: Fraction of time the Frobenius distance of BAHC-filtered matrices is smaller than the alternative estimators. 10,000 independent simulations per point; $t^{out} = 42$ days, $n = 100$ assets, US equities.

Our method both filters out estimation noise and improves the stability of the eigenvectors in a dynamical context. Indeed, the spectral decomposition of BAHC-filtered correlation matrices is close to the optimal CV method with respect to the eigenvalue distribution. Furthermore, in the dynamical context investigated here, the eigenvectors produced by our method have a higher overlap with the out-of-sample ones than the unfiltered in-sample eigenvectors for reasonably small $q = n/t$. This is why our method leads to better minimum-variance portfolios than all the competing filtering methods when the calibration window is small. In particular, if no short selling is allowed, our approach produces, on average, the lowest-risk portfolio.

Future work is needed to characterize the average dependence structure produced by BAHC better, from both theoretical and empirical points of view. In addition, BAHC may still be too strict in some cases and thus leave out valuable information, hence, further refinements of the ansatz will need to be investigated.

## Supporting information

**S1 Appendix.**
(PDF)

**S1 File. Financial dataset code.**
(ZIP)

## Acknowledgments

This publication stems from a partnership between CentraleSupélec and BNP Paribas. This work was performed using HPC resources from the "Mésocentre" computing center of CentraleSupélec and École Normale Supérieure Paris-Saclay supported by CNRS and Région Île-de-France.

## Author Contributions

**Conceptualization:** Christian Bongiorno, Damien Challet.

**Data curation:** Christian Bongiorno, Damien Challet.

**Formal analysis:** Christian Bongiorno, Damien Challet.

**Funding acquisition:** Damien Challet.

**Investigation:** Christian Bongiorno, Damien Challet.

**Methodology:** Christian Bongiorno, Damien Challet.

**Project administration:** Damien Challet.

**Resources:** Damien Challet.

**Software:** Christian Bongiorno, Damien Challet.

**Supervision:** Damien Challet.

**Validation:** Christian Bongiorno, Damien Challet.

**Visualization:** Christian Bongiorno, Damien Challet.

**Writing – original draft:** Christian Bongiorno, Damien Challet.

**Writing – review & editing:** Christian Bongiorno, Damien Challet.

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
