## [Decision Letter · Decision Letter 0]

2 Nov 2020

PONE-D-20-25768

Covariance matrix filtering with bootstrapped hierarchies

PLOS ONE

Dear Dr. Bongiorno,

Thank you for submitting your manuscript to PLOS ONE. After careful consideration, we feel that it has merit but does not fully meet PLOS ONE’s publication criteria as it currently stands. Therefore, we invite you to submit a revised version of the manuscript that addresses the points raised during the review process.

The Reviewers find the research sound, but suggest to clarify some technical aspects as well as the rationale behind the applications. I agree with them that a revision along these lines would improve the paper, making it suitable for publication. 

We look forward to receiving your revised manuscript.

Kind regards,

Prof. Roberta Sinatra

Academic Editor

PLOS ONE

Journal Requirements:

Reviewers' comments:

Reviewer's Responses to Questions

**Comments to the Author**

1. Is the manuscript technically sound, and do the data support the conclusions?

Reviewer #1: Yes

Reviewer #2: Yes

2. Has the statistical analysis been performed appropriately and rigorously? 

Reviewer #1: Yes

Reviewer #2: Yes

3. Have the authors made all data underlying the findings in their manuscript fully available?

Reviewer #1: No

Reviewer #2: Yes

4. Is the manuscript presented in an intelligible fashion and written in standard English?

Reviewer #1: Yes

Reviewer #2: Yes

5. Review Comments to the Author

Reviewer #1: This paper is very interesting and possibly suggest a real progress in the methods to clean noisy covariance matrices. The paper is rather well written, but I have 3 minor suggestions

1) The introduction could be made more crisp. In particular, two paragraphs seem to repeat themselves somewhat: one starting with "Here, we introduce a more flexible hierarchical ansatz able to capture more of the structure of the eigenvectors." and then later "Here, we propose a method that improves on hierarchical clustering. We exploit the fact that the less adequate a hierarchical ansatz, the more fragile it is with respect to small data perturbations. "

2) The Hierarchical Clustering Method cannot be understood on the basis of the information given in the SI, so maybe the authors could improve their explanation in order to make the paper more self contained.

3) The authors may be interested to have a look at the following paper, where the problem of comparing in-sample and out-of-sample eigenvectors is discussed:

Bun, J., Bouchaud, J. P., & Potters, M. (2018). Overlaps between eigenvectors of correlated random matrices. Physical Review E, 98(5), 052145.

Reviewer #2: The paper proposes a bootstrap based method to hierarchically cluster data. The proposed method is termed Bootstrapped Average Hierarchical Clustering (BAHC) and is applied to biological and financial data. A through comparison with competing methods is performed (especially for the financial case, where a real life application is proposed). Clearly the literature on hierarchical clustering is huge and it is very difficult to say whether there are other methods to be used as benchmark.

Overall the paper is interesting and the results are sound. I recommend revision and resubmission, asking authors to respond my comments/criticisms below.

Major remarks:

1) The use of bootstrap in hierarchical clustering has been pioneered in Tumminello et al., Spanning Trees and bootstrap reliability estimation in correlation based networks, International Journal of Bifurcation and Chaos 17, 2319-2329 (2007). Actually the type of bootstrap looks the same, the main difference being that in Tumminello et al authors considers the Minimum Spanning Tree associated with a hierarchical clustering, while the current paper considers the hierarchical tree. A discussion about the similarity between the two approaches should be added and of course the above paper quoted in the bibliography.

2) The method uses a novel clustering algorithm (HCAL of Ref. [9]) to build the filtered matrices C^{(b)<} whose average over bootstrap replicas gives the filtered correlation matrix C^{BACH}. The first obvious question is how critical is the use of HCAL with respect to the many other existing clustering algorithms. Why do authors choose this method? What happens if other methods (such as average linkage) is used. The second question is whether C^{(b)<} is a correlation matrix and if this is important. For example, other clustering methods might provide filtered matrices which are not correlation matrices (for example they are not definite positive), but maybe at the end this is not so important.

3) I am very confused from the biological example. I do not understand what is plotted in Fig. 3 and I am not sure if the displayed result is a good or a bad news for the method. Since Plos ONE is a multidisciplinary journal, it would be nice to have a clearer explanation of this important example.

4) In the portfolio exercise, how do authors estimate past volatilities? Are these simply the standard deviation of returns? The question is relevant since the optimal portfolio strongly depends on the volatilities. Some authors, in order to focus on the estimator of the correlation matrix, use future volatilities. Is this done by the authors?

Minor remarks:

1) Line 176: Authors define q=n/t in the portfolio analysis. However, since the seminal paper by Laloux et al (Phys. Rev. Lett. 1999) the parameter has been defined as Q=T/N (with Laloux's notation). I think it would be less confusing for readers experienced with this type of literature ti use Laloux's Q.

2) Line 185 "linewith" -> "line with"

6. PLOS authors have the option to publish the peer review history of their article (what does this mean?). If published, this will include your full peer review and any attached files.

Reviewer #1: No

Reviewer #2: No

---

## [Author Response · Author response to Decision Letter 0]

25 Nov 2020

Both reviews helped us to clarify and expand our submission where needed. We are grateful to them for this.

About the Data Availability concern, unfortunately, we are not allowed to share the data. However, since they are free downloadable from Yahoo Finance, we included the list of equities and the code to download them in the S.I. 

Reviewer #1: This paper is very interesting and possibly suggest a real progress in the methods to clean noisy covariance matrices. The paper is rather well written, but I have 3 minor suggestions

1) The introduction could be made more crisp. In particular, two paragraphs seem to repeat themselves somewhat: one starting with "Here, we introduce a more flexible hierarchical ansatz able to capture more of the structure of the eigenvectors." and then later "Here, we propose a method that improves on hierarchical clustering. We exploit the fact that the less adequate a hierarchical ansatz, the more fragile it is with respect to small data perturbations. "

Thank you for pointing this out. We have modified the introduction accordingly.

2) The Hierarchical Clustering Method cannot be understood on the basis of the information given in the SI, so maybe the authors could improve their explanation in order to make the paper more self contained.

We have modified this part by improving the text and adding an algorithmic description of the algorithm, including that of the average linkage filtering.

3) The authors may be interested to have a look at the following paper, where the problem of comparing in-sample and out-of-sample eigenvectors is discussed:

Bun, J., Bouchaud, J. P., & Potters, M. (2018). Overlaps between eigenvectors of correlated random matrices. Physical Review E, 98(5), 052145.

We agree with the referee that this reference is relevant to our work, and we included a citation in the revised version. Indeed, prior to the first submission, we analyzed the overlap metric defined in that work. 

We include here an example for N=100,T=150 of the overlap matrix computed over 1000 randomly sampled consecutive windows. Although the results seem promising we are not confident in the interpretation since the marginal distribution of the eigenvalues is extremely different for our method. Furthermore, we are more interested in the N>T regime, whereas that paper investigates the opposite regime. For this reason, we prefer to not include this result in our paper. 

Reviewer #2: The paper proposes a bootstrap based method to hierarchically cluster data. The proposed method is termed Bootstrapped Average Hierarchical Clustering (BAHC) and is applied to biological and financial data. A through comparison with competing methods is performed (especially for the financial case, where a real life application is proposed). Clearly the literature on hierarchical clustering is huge and it is very difficult to say whether there are other methods to be used as benchmark.

Overall the paper is interesting and the results are sound. I recommend revision and resubmission, asking authors to respond my comments/criticisms below.

Major remarks:

1) The use of bootstrap in hierarchical clustering has been pioneered in Tumminello et al., Spanning Trees and bootstrap reliability estimation in correlation based networks, International Journal of Bifurcation and Chaos 17, 2319-2329 (2007). Actually the type of bootstrap looks the same, the main difference being that in Tumminello et al authors considers the Minimum Spanning Tree associated with a hierarchical clustering, while the current paper considers the hierarchical tree. A discussion about the similarity between the two approaches should be added and of course the above paper quoted in the bibliography.

We thank the referee to raise this point. The type of bootstrap is definitely the same. The main difference is that instead of using the bootstraps to associate a reliability value to the clades of the sample dendrogram (or link of the MST as in the cited paper), we totally discard the sample dendrogram, and we consider multiple dendrogram realizations to build or correlation estimator. 

Nevertheless, we agree that this paper was pioneering about this concept of bootstrapping dendrograms, and we included a citation in the text. 

2) The method uses a novel clustering algorithm (HCAL of Ref. [9]) to build the filtered matrices C^{(b)<} whose average over bootstrap replicas gives the filtered correlation matrix C^{BACH}. The first obvious question is how critical is the use of HCAL with respect to the many other existing clustering algorithms. Why do authors choose this method? What happens if other methods (such as average linkage) is used. The second question is whether C^{(b)<} is a correlation matrix and if this is important. For example, other clustering methods might provide filtered matrices which are not correlation matrices (for example they are not definite positive), but maybe at the end this is not so important.

We thank the referee for having noticed such an embarrassing error. C^{(b)<} and C^{<} are not based on Bongiorno et al 2019 (now [11]), rather than on Tumminello et al 2007. Although the citations of the method section are correct, the introduction swapped them.

To the point, the only difference with respect to the method defined in Tumminello et al 2007 was to consider all the clades of the dendrogram. However, this approach is not a novelty of our paper since was already applied by the same authors on Pantaleo et al 2011 (now Ref [9]). 

We agree with the referee that this method can be applied also to non-positive matrices, we believe that this is interesting and it will be investigated in future works. 

3) I am very confused from the biological example. I do not understand what is plotted in Fig. 3 and I am not sure if the displayed result is a good or a bad news for the method. Since Plos ONE is a multidisciplinary journal, it would be nice to have a clearer explanation of this important example.

We thank the referee for this feedback. The point of Fig.3 is to show that a single dendrogram can fail to capture relevant information; therefore, a multi-dendrogram description, proposed in this work, should be preferred. In particular, we have shown that different bootstrap realizations of an HC dendrogram, instead of being scattered around a central unique dendrogram, they cluster around two or more centroid dendrograms. So this is bad news for the strict HC. 

We extended the microarray section to clarify this concept. 

4) In the portfolio exercise, how do authors estimate past volatilities? Are these simply the standard deviation of returns? The question is relevant since the optimal portfolio strongly depends on the volatilities. Some authors, in order to focus on the estimator of the correlation matrix, use future volatilities. Is this done by the authors?

We thank the referee for this excellent idea. The volatility estimator used in this paper is the historical standard deviations, we clarified that in the main text. However, we agree with the referee that considering future realized volatility can provide an upper bound of the performances of the method. We’ve included in SI such analysis, but we did not observe substantial qualitative differences. 

Minor remarks:

1) Line 176: Authors define q=n/t in the portfolio analysis. However, since the seminal paper by Laloux et al (Phys. Rev. Lett. 1999) the parameter has been defined as Q=T/N (with Laloux's notation). I think it would be less confusing for readers experienced with this type of literature to use Laloux's Q.

We thank the referee for this observation. However, we think that is not possible to avoid confusion in the reader about that since other authors define q=n/t. For example in the recent review 

Bun, J., Bouchaud, J. P., & Potters, M. (2017). Cleaning large correlation matrices: tools from random matrix theory. Physics Reports, 666, 1-109.

On page 4, they use q=n/t. Since we believe that actually, recent works by Bouchaud and coworkers are more relevant to this topic, we prefer to leave this definition of q. 

2) Line 185 "linewith" -> "line with"

Corrected, thanks.

---

## [Decision Letter · Decision Letter 1]

22 Dec 2020

Covariance matrix filtering with bootstrapped hierarchies

PONE-D-20-25768R1

Dear Dr. Bongiorno,

We’re pleased to inform you that your manuscript has been judged scientifically suitable for publication and will be formally accepted for publication once it meets all outstanding technical requirements.

Kind regards,

Roberta Sinatra

Academic Editor

PLOS ONE

Additional Editor Comments (optional):

All Reviewers agree that the revised manuscript addresses all comments from the first round of review and and they recommend it for publication. I agree with them.

Reviewers' comments:

Reviewer's Responses to Questions

**Comments to the Author**

1. If the authors have adequately addressed your comments raised in a previous round of review and you feel that this manuscript is now acceptable for publication, you may indicate that here to bypass the “Comments to the Author” section, enter your conflict of interest statement in the “Confidential to Editor” section, and submit your "Accept" recommendation.

Reviewer #1: All comments have been addressed

Reviewer #2: All comments have been addressed

2. Is the manuscript technically sound, and do the data support the conclusions?

Reviewer #1: Yes

Reviewer #2: Yes

3. Has the statistical analysis been performed appropriately and rigorously? 

Reviewer #1: Yes

Reviewer #2: Yes

4. Have the authors made all data underlying the findings in their manuscript fully available?

Reviewer #1: Yes

Reviewer #2: Yes

5. Is the manuscript presented in an intelligible fashion and written in standard English?

Reviewer #1: Yes

Reviewer #2: Yes

6. Review Comments to the Author

Reviewer #1: no additional comments, all remarks have been addressed. Authors have explained why the data cannot be made fully available

Reviewer #2: The revised version fully answers my previous comments. I recommend it for publication on PlosOne.

7. PLOS authors have the option to publish the peer review history of their article (what does this mean?). If published, this will include your full peer review and any attached files.

Reviewer #1: No

Reviewer #2: No

---

## [Editor Report · Acceptance letter]

5 Jan 2021

PONE-D-20-25768R1 

Covariance matrix filtering with bootstrapped hierarchies 

Dear Dr. Bongiorno:

I'm pleased to inform you that your manuscript has been deemed suitable for publication in PLOS ONE. Congratulations! Your manuscript is now with our production department. 

Kind regards, 

on behalf of

Prof. Roberta Sinatra 

Academic Editor

PLOS ONE